# A Subject-Specific Approach to Detect Fatigue-Related Changes in Spine Motion Using Wearable Sensors

**DOI:** 10.3390/s20092646

**Published:** 2020-05-06

**Authors:** Victor C.H. Chan, Shawn M. Beaudette, Kenneth B. Smale, Kristen H.E. Beange, Ryan B. Graham

**Affiliations:** 1School of Human Kinetics, University of Ottawa, 200 Lees Avenue, Ottawa, ON K1S 5L5, Canada; vchan017@uottawa.ca (V.C.H.C.); sbeaudette@brocku.ca (S.M.B.); kenneth.smale@angels.com (K.B.S.); kristenbeange@cmail.carleton.ca (K.H.E.B.); 2Department of Kinesiology, Brock University, 1812 Sir Isaac Brock Way, St. Catharines, ON L2S 3A1, Canada; 3Department of Systems and Computer Engineering, Carleton University, 1125 Colonel by Drive, Ottawa, ON K1S 5B6, Canada

**Keywords:** muscle fatigue, inertial measurement units, composite index, subject-specific, spine

## Abstract

An objective method to detect muscle fatigue-related kinematic changes may reduce workplace injuries. However, heterogeneous responses to muscle fatigue suggest that subject-specific analyses are necessary. The objectives of this study were to: (1) determine if wearable inertial measurement units (IMUs) could be used in conjunction with a spine motion composite index (SMCI) to quantify subject-specific changes in spine kinematics during a repetitive spine flexion-extension (FE) task; and (2) determine if the SMCI was correlated with measures of global trunk muscle fatigue. Spine kinematics were measured using wearable IMUs in 10 healthy adults during a baseline set followed by 10 sets of 50 spine FE repetitions. After each set, two fatigue measures were collected: perceived level of fatigue using a visual analogue scale (VAS), and maximal lift strength. SMCIs incorporating 10 kinematic variables from 2 IMUs (pelvis and T8 vertebrae) were calculated and used to quantify subject-specific changes in movement. A main effect of set was observed (*F* (1.7, 15.32) = 10.42, *p* = 0.002), where the SMCI became significantly greater than set 1 starting at set 4. Significant correlations were observed between the SMCI and both fatigue VAS and maximal lift strength at the individual and study level. These findings support the use of wearable IMUs to detect subject-specific changes in spine motion associated with muscle fatigue.

## 1. Introduction

Muscle fatigue is known to reduce the force generation capacity of a muscle [1,2], alter movement coordination [3,4,5,6], and increase an individual’s risk for musculoskeletal (MSK) injury [6,7,8]. Despite this knowledge, fatigue-related injuries are still common in workplace and sport settings [8,9,10,11], with lower back disorders (LBDs) being the most common and costly MSK injury [12,13,14]. This suggests that a reliable, objective method for tracking fatigue may be beneficial to reduce fatigue-related injuries in work and sport settings. However, tracking fatigue is difficult, as current methods are limited to individual subjective appraisals, such as the visual analogue scale (VAS) [15,16], or traditional objective laboratory-based measures like strength assessments [17] and electromyography [18]. With the advent of wearable inertial measurement units (IMUs), human movement (kinematic) data can now be gathered in greater quantities and in a variety of environments with acceptable spatial and temporal resolutions. These kinematic data may provide insight into fatigue status; however, individuals have been shown to demonstrate heterogeneous kinematic responses to fatigue [19,20] warranting subject-specific methods of analysis [21,22]. The ability of wearable IMUs to detect subject-specific alterations in movement patterns have been explored in walking and running gait and show promise in their ability to detect kinematic changes that are associated with fatigue [21,22].

A proposed method to track subject-specific alterations in movement kinematics involves the implementation and tracking of a composite index that incorporates multiple relevant kinematic variables [21,23]. Individuals’ “typical” movement patterns during an activity or task of interest are quantified during an unfatigued (i.e., baseline) set using selected variables; then, composite indices can be computed for subsequent repetitions/sets by comparing these repetitions/sets to the individuals’ own typical movement patterns. This allows changes in an individual’s movements to be quantified in terms of standard deviations (SD) away from their typical movement. If changes in muscle fatigue are found to be correlated with changes in the composite index, then tracking the changes in composite indices may be a promising method for objectively identifying when fatigue occurs. Variables that comprise the composite index should be selected as being relevant to the phenomenon of interest (e.g., muscle fatigue) while also being available at a reasonable computational cost [23].

An example of a relevant variable that can be quantified with low computational cost for use in a movement composite index is continuous relative phase (CRP). CRP is used to investigate movement coordination and is quantified as the difference in phase angle between two adjacent segments in oscillation, derived from the phase plane of the segments [24]. CRP has been employed extensively in spine control research because it can differentiate between normal and abnormal spine movement [25,26,27], detect individual spine movement subtypes [28], reflect changes in muscle fatigue status [29,30], and be measured reliably using wearable IMUs [31]. In this study, 10 variables in the sagittal plane were selected to comprise a spine motion composite index (SMCI) for their known association with muscle fatigue and/or low computational processing cost: peak value of the thoraco-pelvic CRP waveform; repetition time; and IMU (pelvis and T8 vertebrae) orientation range, peak orientation, angular velocity, and angular acceleration.

The primary objective of this study was to determine if wearable IMUs used with an SMCI could quantify subject-specific changes in spine kinematics during a repetitive flexion-extension (FE) task. The secondary objective of this study was to determine if the observed changes in SMCI are correlated to changes in global trunk muscle fatigue, quantified using fatigue VAS and maximal lift strength assessments. It was hypothesized that subject-specific changes in spine kinematics throughout the sets could be quantified using wearable IMUs and an SMCI, and that changes in the SMCI would be significantly correlated to fatigue measures.

## 2. Methods

### 2.1. Participants

Ten volunteer participants (5 male and 5 female) were recruited from the Ottawa area if they met the following inclusion criteria: aged 19 years or older; no history of low back pain in the past six months; and no history of MSK injuries in the past six months. The study procedure was approved by the institutional Research Ethics Board, which required all participants to provide informed consent prior to data collection. Participants were also asked to complete a Baecke Questionnaire for Measurement of a Person’s Habitual Physical Activity [32].

### 2.2. Instrumentation

Data were collected using the Xsens MVN Link inertial motion capture system (Xsens Technologies B.V., Enschede, Netherlands). Seventeen IMUs were secured to participants using an Xsens MVN Link Lycra Suit in the following positions: one on the head, T8 vertebrae, and pelvis; and bilaterally on the shoulders, upper arms, forearms, hands, thighs, shanks, and feet [33]. Three-dimensional (3D) gyroscope, accelerometer, and magnetometer data were sampled at 240 Hz and transmitted via Xsens MVN Link Access Point to Xsens MVN Analyze software (Xsens Technologies B.V., Enschede, Netherlands). Sensor fusion algorithms that estimate body segment position and orientation from raw sensor data are implemented in the software [33,34], which have been previously validated against optoelectronic motion capture systems [34,35,36].

The maximal isometric lift strength assessments (described in Section 2.3) were conducted using an s-type load cell with a handle attached and adjusted to participants’ knee-level (Figure 1c). These data were sampled at 100 Hz and collected using custom LabVIEW software (National Instruments, Austin, TX, USA).

### 2.3. Fatiguing Protocol

Emulating previous movement protocols [31,37,38,39], participants were constrained at the hip and instructed to touch two targets with their arms outstretched in synchrony with a metronome set at 0.5 Hz (i.e., 4 s per FE repetition) for the spine FE task. Both targets were placed directly anterior to the participants: the first target at shoulder-level and at arms-length away (Figure 1a), and the second target at knee-level, 50 cm anterior to the knees (Figure 1b). One FE repetition was defined as movement from the upright position (Figure 1a) into the flexed position (Figure 1b) and back to the upright position (Figure 1a).

Participants performed one baseline set of 50 spine FE repetitions followed by an eight-minute rest. Then, participants performed ten fatiguing sets of 50 spine FE repetitions with no rest between sets. Immediately after each set (including baseline), participants’ global trunk muscle fatigue was assessed using two methods: a self-report of their perceived level of fatigue using a ten-centimeter VAS (measured in millimeters; mm) [15,16]; and a maximal isometric lift strength assessment of the maximal tensile force that could be exerted on the load cell (by pulling upwards on the handle; Figure 1c) [17]. This assessment was based off a similar protocol described in [17]. Participants were instructed to grasp the handle using both arms and to perform three consecutive maximal exertions per assessment. Participants were constrained at the hip and instructed to keep their legs straight while extending their spine to “ramp up” the amount of force they exerted until their maximum force was reached. Participants were asked to complete all FE sets and fatigue assessments unless they were too fatigued to continue, or their maximal lift strength fell below 70% of their baseline value [40].

### 2.4. Data Processing

Fused sensor orientation data (in quaternions) from the 17 IMU sensors were extracted using the Xsens MVN Analyze data collection software. However, only the sensor orientation quaternions from the pelvis and T8 vertebrae were used in this study. These data were exported to MATLAB R2019b (MathWorks, Natick, MA, USA), where they were converted to Euler angles and smoothed using a zero-lag (effective 4th order) Butterworth low-pass filter [41] with a cut-off frequency of 3Hz.

Sensor orientations were used to calculate the relative motion between the pelvis and thorax segments (i.e., spine motion) using a flexion-extension/lateral-bend/axial-twist rotation sequence. Angular velocities and accelerations of the individual pelvis and T8 sensors were also calculated using the techniques described in [41]. The troughs of the sagittal component of the spine motion signal (corresponding to the position shown in Figure 1a) were used to locate the start and end of the FE repetitions. The first ten FE repetitions of each set were excluded from analyses to ensure a steady-state motion was achieved [31,38,42].

Fatigue VAS was quantified to the nearest 0.1 mm by a researcher who measured the distance between the zero-line and the mark made by participants on the paper using a ruler. The average of the three peak values obtained during each maximal lift strength assessment was calculated, resulting in one value/set (i.e., eleven values/participant).

### 2.5. Data Analysis

Ten variables in the sagittal plane were selected to comprise an SMCI: peak value of the thoraco-pelvic CRP waveform, repetition time, and IMU (pelvis and T8 vertebrae) orientation range, peak orientation, angular velocity, and angular acceleration. These variables were selected to fulfill the following criteria: (1) known association with muscle fatigue and/or LBD injury risk [29,43,44,45]; and (2) low computational demand. Variables are described below.

#### 2.5.1. Continuous Relative Phase

To produce the thoraco-pelvic CRP waveform, the sagittal orientations and angular velocities of the pelvis and thorax segments were divided into separate FE repetitions and time-normalized to 101 samples to represent 0–100% of the FE repetition. The sagittal orientations (*θ*) and angular velocities (ω) of all repetitions and sets were then independently phase-normalized from −1 to +1 using Equation (1) [46,47]:(1)$$\theta_{i,norm}=2\times \frac{\theta_{i}-\min\left(\theta \right)}{\max\left(\theta \right)-\min\left(\theta \right)}-1$$

Phase portraits of normalized orientation plotted against normalized angular velocities were created [31,46,48]. Phase angles (PAs) were then calculated using a four-quadrant inverse tangent function, producing values that range from −180° to +180° (with respect to the positive *X*-axis) [25,31,49]. The CRP waveform was then determined using Equation (2) [31,47,48,49]:(2)$$CRP_{i}=\left|\varphi_{i,pelvis}-\varphi_{i,thorax}\right|$$
where *φ_pelvis_* is the pelvis PA, and *φ_thorax_* is the thorax PA. The peak value of the CRP waveform was identified for each repetition and was used in the SMCI over other relevant variables derived from CRP (e.g., deviation phase) because it can be quantified for each repetition. Mean and SD of peak CRP were calculated for the baseline set of each participant. The peak CRP of each FE repetition during the 10 fatiguing sets was recorded and used to calculate the SMCI.

#### 2.5.2. Repetition Time

The time between the start and end of each FE repetition was used to represent repetition time. Although this was constrained using a metronome, individuals may have still shown repetition-to-repetition variability due to fatigue [45]. Mean and SD of FE repetition time were calculated for the baseline set of each participant. The repetition time of each FE repetition during the 10 fatiguing sets was recorded and used to calculate the SMCI.

#### 2.5.3. Orientation Range, Peak Orientation, Angular Velocity, and Angular Acceleration

The peak orientation, angular velocity, and angular acceleration of the pelvis and T8 sensors in the sagittal plane (i.e., rotation about the mediolateral axis) were identified for every FE repetition. The troughs of the sagittal orientation signal of the pelvis and T8 sensors were also identified for every FE repetition; this minimum value was subtracted from the corresponding peak value to produce the orientation range for every FE repetition. Mean and SD of pelvis and T8 sensor orientation range, peak orientation, peak angular velocity, and peak angular acceleration in each set were calculated for the baseline set of each participant; then, these 8 variables were quantified for each FE repetition during the 10 fatiguing sets and used to calculate the SMCI.

#### 2.5.4. Spine Motion Composite Index

The mean (*μ_Typical_*) and SD (*σ_Typical_*) for each of the 10 variables from participants’ baseline set were used to establish their “typical” movement. Then, the SMCI (*z*) was calculated for each FE repetition using the value of each variable in the FE repetition (*x*) with the following equation [21]:(3)$$z_{i}=\frac{1}{10}\sum_{j=1}^{10}\frac{\left|x_{ij}-{\left(\mu_{Typical}\right)}_{j}\right|}{{\left(\sigma_{Typical}\right)}_{j}},\;i=\mathrm{each}\;\mathrm{FE}\;\mathrm{repetition},\;j=\mathrm{each}\;\mathrm{variable}$$

Forty SMCIs per fatiguing set (one per FE repetition) per participant were calculated using Equation (3). The average of the 40 SMCIs was calculated to obtain one SMCI per fatiguing set, per participant. The SMCI represents the SDs above/below the mean of the participants typical, unfatigued spine movement measured during their baseline set.

#### 2.5.5. Statistical Analyses

All statistical analyses were performed using R version 3.6.1 (R Core Team, Vienna, Austria) with the significance level for all tests set to *α* = 0.05. The dependent variables (DV) of the study were the SMCI, fatigue VAS, and maximal lift strength. Normality for each DV was assessed using the Shapiro-Wilk test [50]. To address the primary purpose, a one-way, repeated-measures analysis of variance (ANOVA) was performed to determine the effect of fatiguing set on the SMCI. *F*-ratio degrees of freedom (DF) were corrected using the Greenhouse-Geisser є value if sphericity (assessed using Mauchly’s test of sphericity) was violated. If a main effect of set existed, *post hoc* analyses using Bonferroni-corrected paired, one-tailed Student’s *t*-tests were conducted for the SMCI between the first set and sets 2–10 (Bonferroni *α*_adjusted_ = 0.05/9 = 0.006). These comparisons would indicate which sets participants’ SMCI were significantly greater than their first set (baseline comparisons were not possible as the SMCI is available at starting at set 1).

To address the secondary purpose, correlations between the SMCI and the fatigue measures (fatigue VAS and maximal lift strength) were calculated using three methods: (1) Pearson’s correlation coefficient (*r*) for each participant; (2) repeated-measures correlation coefficient (*r*_rm_) for all participants; and (3) Pearson’s correlation coefficient (*r*) for the DVs averaged across all participants. Method (1) was employed to investigate the subject-specific correlations between the SMCI and fatigue measures; method (2) was employed to determine if intra-individual correlations between DVs are homogenous between individuals [51]; and method (3) was performed to determine the association between the SMCI and the fatigue measures overall (i.e., across all participants). For all methods, a correlation of 0.10–0.29, 0.30–0.49, and 0.5+ were interpreted as weak, moderate, and strong, respectively [51,52].

## 3. Results

### 3.1. Participants

Participant demographic characteristics are presented in Table 1. No participants withdrew from the fatiguing protocol nor were any asked to stop before the completion of the study. From baseline to set 10, an average increase of 39.4 mm (348.7%) in fatigue VAS and an average decrease of 129.9 N (19.7%) in maximal lift strength were observed.

### 3.2. Influence of Fatiguing Set on Spine Motion Composite Index

The results of the Shapiro-Wilk tests revealed that the distribution for the SMCI, fatigue VAS, and maximal lift strength were not significantly different from a normal distribution (*p* ≥ 0.08). Mauchly’s test revealed that SMCI had violated sphericity; thus, *F*-ratio DF were corrected using Greenhouse-Geisser є. For the SMCI, a significant main effect of set was observed (*F* (1.7, 15.32) = 10.42, *p* = 0.002). *Post hoc* analyses revealed that the SMCI was significantly greater in sets 4–10 compared to set 1 (*p* ≤ 0.003), but not in sets 2 and 3 (*p* ≥ 0.013). The results are presented in Figure 2.

### 3.3. Correlation Between Spine Motion Composite Index and Fatigue Measures

The subject-specific Pearson’s correlation coefficients are presented in Table 2 and Figure 3. A significant correlation was observed between the SMCI and at least one fatigue measure for 8 of 10 participants. Specifically, a strong correlation between the SMCI and fatigue VAS was observed for 7 participants (6 positive and 1 negative correlation), and between the SMCI and maximal lift strength for 5 participants (4 negative and 1 positive correlation).

The results of the repeated measures correlation showed a moderate, positive correlation (*r*_rm_(89) = 0.45, *p* < 0.001) between the SMCI and fatigue VAS. A moderate, negative correlation (*r*_rm_(89) = −0.49, *p* < 0.001) was observed between the SMCI and maximal lift strength.

When DVs were averaged across participants, a strong, positive correlation was observed between the SMCI and fatigue VAS (*r*(8) = 0.92, *p* < 0.001) and a strong, negative correlation was observed between the SMCI and maximal lift strength (*r*(8) = −0.91, *p* < 0.001).

## 4. Discussion

The primary objective of this study was to determine if wearable IMUs used in conjunction with an SMCI could quantify subject-specific changes in spine kinematics during a repetitive FE task. The current findings showed that this instrumentation and an SMCI comprised of 10 spine kinematic variables were sensitive to subject-specific changes that occurred throughout a fatiguing protocol. Beginning at fatiguing set 4, participants (on average) performed the repetitive FE task in a significantly different manner compared to their first fatiguing set. The secondary objective was to determine if observed changes in the SMCI were correlated to changes in fatigue VAS and maximal isometric lift strength. At the individual level, a strong correlation between the SMCI and one or more fatigue measures existed for 8 of 10 participants. Repeated measures correlation analyses showed moderate correlations between the SMCI and the fatigue measures and suggest that the intra-individual associations were moderately heterogeneous between individuals. When overall changes across all participants were considered, the SMCI was strongly correlated to both fatigue measures. Thus, the results support an association between the SMCI measured using wearable IMUs and changes in fatigue VAS and maximal lift strength.

This novel method of using wearable IMUs and a composite index to quantify subject-specific typical movement was developed for the purpose of tracking running biomechanics [21]. A major strength of this approach is the ability to normalize observations to participants’ own baseline kinematics. To the authors’ knowledge, this was the first study to implement this subject-specific approach to quantify “typical” spine kinematics, and the findings provide support for the use of wearable IMUs and composite indices to detect fatigue-related changes in spine kinematics. As data were collected from two IMUs (i.e., pelvis and T8 vertebrae) and required low computational cost to calculate the SMCI for individual repetitions, this method of detecting muscle fatigue has potential to be more objective and practical for quantifying fatigue level in work and sport settings compared to traditional subjective appraisals, strength assessments, or electromyography. The affordability and ease of use of wearable IMUs also allow for the development of an inexpensive, real-time monitoring system that incorporates the use of these devices with mobile applications and cloud computing [53]. Such systems can alert workers, athletes, supervisors, or coaches when their movement is becoming significantly atypical and indicative of muscle fatigue. Thus, wearable IMUs used with subject-specific composite indices have potential to help mitigate fatigue-related MSK injuries in work and sport by optimizing subject-specific work-rest ratios [8].

The current findings show that the SMCI changes throughout the fatiguing protocol and that this method was able to detect significantly atypical spine motion beginning at set 4 (compared to the first fatiguing set). At set 4, fatigue VAS had increased by 21.8 mm (192.9%) and maximal lift strength had decreased by 55.6 N (8.8%) on average compared to baseline. It is unlikely that such small decrements in strength can indicate the occurrence of muscle fatigue [40]; these results show that a subject-specific method may be sensitive enough to detect early stages of muscle fatigue development by observing changes in spine motion alone. Being able to objectively detect fatigue-related changes in spine kinematics before an individual is fully fatigued is important to attenuate the related MSK injury risk [8]. That is, this method may not be practical if it is only able to detect changes in spine kinematics after individuals were fully fatigued. The early detection of subject-specific kinematic changes is helpful to monitor muscle fatigue status if these kinematic changes are correlated to their perceived muscle fatigue or changes in maximal force production.

Strong correlations were found between the changes in SMCI and the fatigue measures at the individual and study levels. The presence of these associations may be attributable to the fact that some variables in the SMCI were significantly correlated to the fatigue measures on their own (correlation heat maps for each variable are presented in Appendix A; Figure A1), and that these variables have been previously linked to muscle fatigue. For example, research has shown that global trunk muscle fatigue was associated with an increase in peak spine flexion angle during a FE task [29]. Furthermore, Hu and Ning (2015) demonstrated that after a trunk muscle fatiguing protocol, spine coordination and variability (derived using CRP) significantly decreased during a lifting task [30]. Aside from their computational simplicity, some of the kinematic variables included in the SMCI (e.g., T8 vertebrae sensor peak orientation) may have contributed to the association with the fatigue measures because these variables are used in the calculation of other complex features that have been associated with global trunk muscle fatigue (e.g., spine local dynamic stability) [5,54].

Changes in the SMCI were not associated with either muscle fatigue measure for 2 individuals. A possible explanation is that these individuals may have only experienced low levels of fatigue: S02 reported less subjective fatigue at the end of the protocol compared to baseline; whereas S10 showed a 17.5% decrease in maximal lift strength (individual responses presented in Appendix A; Figure A2). This may suggest that when changes in fatigue status are minimal, a composite index may not reflect subject-specific changes in kinematics. Furthermore, some heterogeneity in the direction of correlation between the SMCI and fatigue measures was observed amongst individuals. This may be a result of diverse kinematic responses to fatigue. For example, research has reported increased and decreased movement variability, coordination, and spine local dynamic stability in response to fatigue [19,55,56]. Future efforts should be directed at determining which kinematic variables are best for inclusion in a composite index on a subject-by-subject basis. Previous work has shown that machine learning algorithms may perform better if feature selection is performed for each individual [57], suggesting that composite indices could also be better tailored to each individual if feature selection (e.g., correlation-based feature selection) [58] was performed. Still, the correlations found in the current study between the SMCI and fatigue measures are significant, supporting its potential to be used for monitoring of muscle fatigue development.

The results of this study should be considered with some limitations. First, the FE movements were simple, repetitive, and constrained to the sagittal plane because the proposed method of quantifying subject-specific typical movement requires the motion to be repetitive and/or cyclical in nature (e.g., running) [21]. Although this method may still be effective for more complex, repetitive movements, a preliminary attempt with a simple, uniplanar movement was warranted to support the use of this novel method to quantify spine kinematics. Future efforts should be directed at implementing this method with more complex tasks, such as repetitive asymmetrical lifting. Second, there were relatively few fatiguing sets in this study, limiting the statistical power available for the subject-specific and overall correlations. As such, it was imperative that individual-level correlations were strong to be detected in this study. Lastly, the sample size for this study was relatively small. Although the 10 variables used in the SMCI revealed subject-specific changes in spine motion and were significantly correlated to participants’ fatigue measures, future efforts should be directed at determining if this remains applicable for a wider variety of participants. Nonetheless, the current findings support the potential of this approach to detect global trunk muscle fatigue as the variables used in a composite index can be tailored to individuals, and because the baseline movement characteristics are defined using subject-specific data (rather than study sample data).

## 5. Conclusions

Wearable IMUs are becoming an accessible tool for collecting kinematic data in various environments and can enable subject-specific analyses. The results of this study show that a novel method employing wearable IMUs with a composite index can be used to detect subject-specific changes in spine motion, and these changes are correlated with subjective and objective measures of global trunk muscle fatigue. Thus, the use of wearable IMUs with composite indices have the potential to detect the onset of muscle fatigue in real-time and mitigate fatigue-related injury risk in workplace and sport settings.

## Figures and Tables

**Figure 1 sensors-20-02646-f001:**
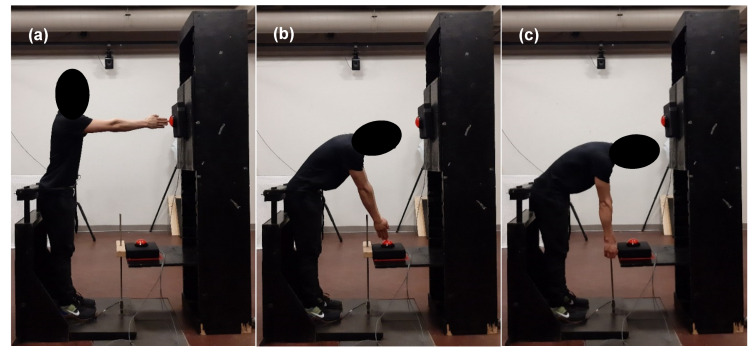
Participants were constrained at the hip for the spine flexion-extension task. To begin, (**a**) participants touched the first target placed at shoulder-level, then (**b**) flexed their spine to touch the second target placed at knee-level, before (**a**) returning to the initial position. For the maximal lift strength assessment, (**c**) participants exerted maximal effort to pull upwards on the load cell handle that was set at knee-level.

**Figure 2 sensors-20-02646-f002:**
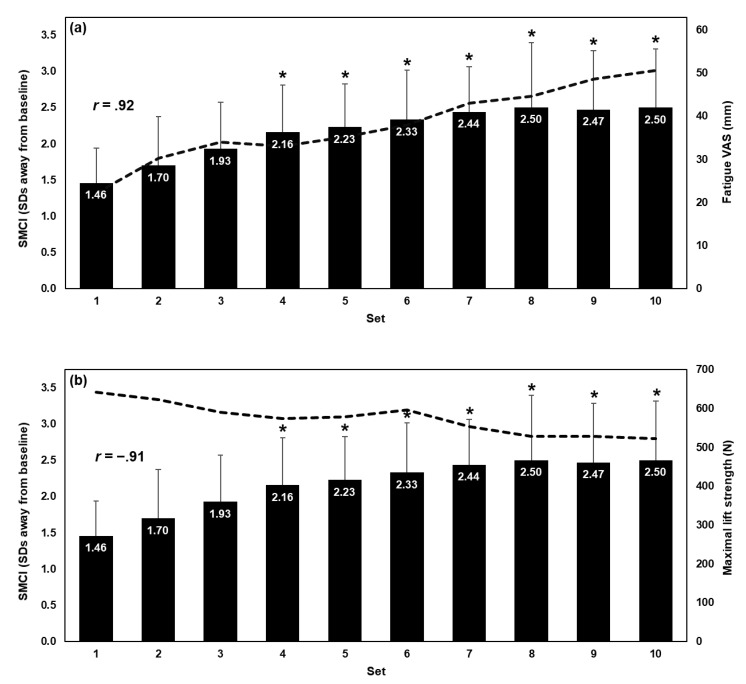
The spine motion composite index (SMCI; bars) plotted with (**a**) fatigue visual analogue scale (VAS; dashed line) and (**b**) and maximal lift strength (dashed line). These data were averaged across participants for fatiguing sets 1–10. Error bars indicate the SMCI standard deviation (SD), and asterisks indicate that the SMCI of that set is significantly greater than set 1. When averaged across participants, strong Pearson’s correlation coefficients (*r*) were observed between the SMCI and fatigue VAS (*r*(8) = 0.92, *p* < 0.001) and between the SMCI and maximal lift strength (*r*(8) = −0.91, *p* < 0.001).

**Figure 3 sensors-20-02646-f003:**
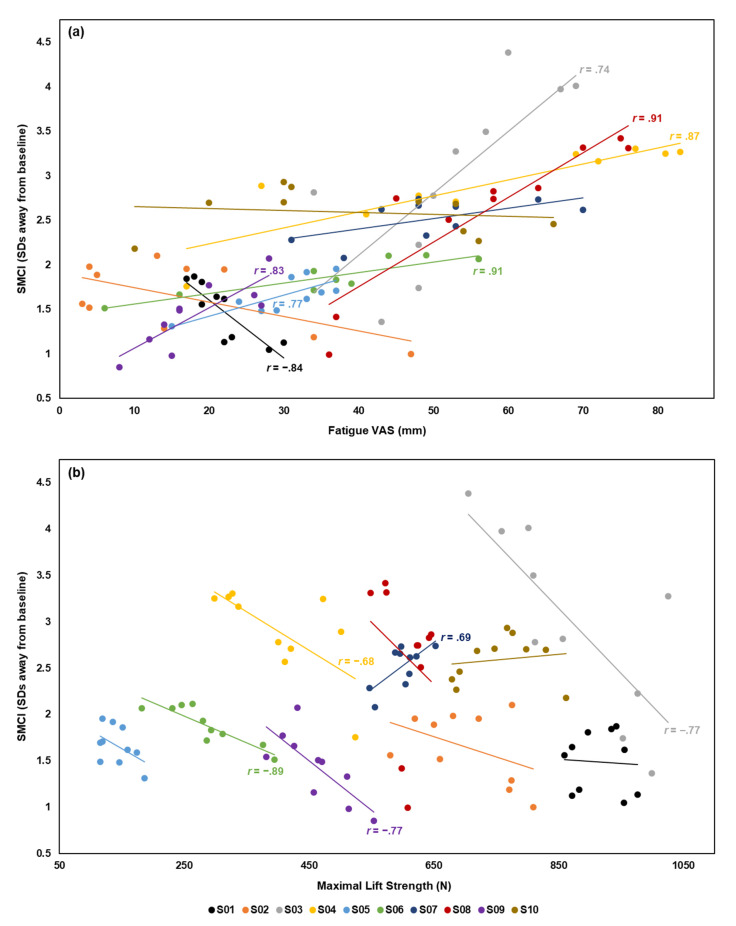
The subject-specific correlations between (**a**) the spine motion composite index (SMCI) and fatigue visual analogue scale (VAS) and between (**b**) the SMCI and maximal lift strength. Participants are identified as S01 to S10. Pearson’s correlation coefficients (*r*) are presented with the subject-specific trendlines for statistically significant correlations (*α* = 0.05).

**Table 1 sensors-20-02646-t001:** Mean (standard deviation) for participant age, mass, and height.

Demographic	Male	Female
N	5	5
Age (years)	29.6 (10.8)	27 (3.3)
Mass (kg)	78.46 (18.55)	59.86 (9.20)
Height (cm)	178.1 (11.3)	173.3 (10.2)

**Table 2 sensors-20-02646-t002:** The subject-specific Pearson’s correlation coefficients (*r*) between the spine motion composite index (SMCI) and fatigue visual analogue scale (VAS), and between the SMCI and maximal lift strength. The degrees of freedom (DF) = 8 for all correlations.

Participant	Fatigue VAS (mm)		Maximal Lift Strength (N)	
	*r*	*p* Value	*r*	*p* Value
S01	−0.84	< 0.001 *	−0.06	0.863
S02	−0.61	0.061	−0.43	0.210
S03	0.74	0.014 *	−0.77	0.009 *
S04	0.87	0.001 *	−0.68	0.029 *
S05	0.77	0.009 *	−0.48	0.156
S06	0.91	< 0.001 *	−0.89	< 0.001 *
S07	0.59	0.072	0.69	0.027 *
S08	0.91	< 0.001 *	−0.27	0.445
S09	0.83	0.003 *	−0.77	0.009 *
S10	−0.15	0.671	0.15	0.677

* indicates significance with *α* = 0.05.

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
