# Peer review of "A Subject-Specific Approach to Detect Fatigue-Related Changes in Spine Motion Using Wearable Sensors"

_sensors, 2020, doi:10.3390/s20092646_

Round 1

Reviewer 1 Report

This manuscript studied the fatigue related changes in spine motion using the wearable IMUs, the idea is novel and very interesting. However, the manuscript is not well presented and the experiments are not very persuaded. The reviewer’s comments are as follows:

1. The reference citation form is not correct, there should be a blank space between two reference numbers. e.g. [19, 20]

2. The fatigue leaded by repetitive flexion-extension can’t represent fatigue leaded by other activities, so the meaning of the research results is very limited; Is the proposed fatigue detecting approach only work for the repetitive spine flexion-extension exercise?

3. The fatigue measures used as references are too outdated, one was published in 1997, the other was published in 1999.

4. The results of this study lack persuasion as the number of participants were too small and only 8 of 10 participants showed significant correlation between the SMCI and fatigue measure.

5. Other existing methods should be compared to prove the advantage of the proposed method.

6. Why the pelvis and the T8 vertebrae were selected to collect fatigue related data? Can the IMU sensors be worn in other body parts?

Reviewer 2 Report

The manuscript proposed an outcome parameter “spine motion composite index”, which was derived from kinematic variables measured by IMU, as an indicator of fatigue in trunk flexion-extension exercises. The indicator was then validated by correlation with a common fatigue assessment method, a questionnaire, and a lifting strength test.

The study produced some impact since the quantification of neuromuscular fatigue is a challenging task to the scientist. Three general queries were as followed:

  1. I would like to know the reason for choosing to study the fatigue of trunk flexion/extension exercise. While prolonged sitting in office settings targeted on postural muscles rather than motor muscles, it is better to suggest a more specific potential application on the scenario of trunk flexion-extension.
  2. I believe that I have a different understanding and interpretation of “subject-specific approach” from the authors. I don’t agree that this is a “positive” terms, while the authors were overwhelming this feature over the text. Subject-specific approach could mean that you need to have different consideration, instrument or algorithm that targets to the individual specific feature, and it often comes across when a doctor makes diagnose to a patient because they may have a different history. Otherwise, it could mean that your study only adopt one representative subject and it is common in some theoretical study such as finite element analysis. Personally, I think it is a negative term since your findings or invention might not be able to generalize to use in all people. In fact, I think your study is advantageous. Actually, your study adopted a method that normalized the index to the baseline kinematic characteristics of the subject, such that your algorithm is indeed universal and not subject-specific. I suggest using some terms, such as a normalized index or personalized.
  3. Similarly, the title is too focused on “subject-specific” that I cannot agree more. It would be better to focus on the contribution, such as proposing some fatigue indicator by an index or instrument, then followed by the approach.

Some specific queries are as followed:

Intro, Line 58: The authors mentioned that the use of CRP because it is computationally less costly. There could be simpler methods to evaluate fatigue. For example in the following reference, simply by the change in kinematics variable or the variability of these variables. I suggested to have a brief review of the simple index and come up with a compromise that CRP could provide more feature that the former could not achieve.

Maas, E., De Bie, J., Vanfleteren, R., Hoogkamer, W. and Vanwanseele, B., 2018. Novice runners show greater changes in kinematics with fatigue compared with competitive runners. Sports biomechanics17(3), pp.350-360.

Wong, D.W.C., Lam, W.K. and Lee, W.C.C., 2020. Gait asymmetry and variability in older adults during long-distance walking: Implications for gait instability. Clinical biomechanics72, pp.37-43.

Method, Line 78: Please include the planned inclusion criteria, for example, the age range, planned number of different gender of participants, and where would be the participants recruited.

Line 87: Any reference on the set of sensor placement? Otherwise, state your empirical reason.

Line 93: Include a brief statement on how the load cell facilitates the maximal lifting test.

Line 100:  “knee level”?

Figure 100: you may separate the in-text referencing of Figure 1a, 1b and 1c.

Line  102: I have some confusion with your “trials” and “cycles”, since the trial is often intuitively defined by readers with a single run. You may want to use the terms “reps” and “sets”, which was commonly used to describe repetitions in physical exercise protocols. Anyhow, the definition of reps and sets need to be declared and referenced.

Line 106: I feel confused with the maximal lifting test. Did the participant require to maintain the posture? In Figure 1c, It looks like that participant was pushing on the handle. It would be better to add some arrows to indicate the motion direction and also highlight the position of your load cell.

Line 159: Please make sure the naming of duty cycle is correct. It normally means a time ratio.

Results, Line 213, Naming the sub-titles using a statistical test is not very meaningful. I suggested to rename it as “the influence of exercise cycles on SMCI and fatigue test outcome” , and “correlation between SMCI and fatigue test outcome”, or anything that is more accurate and meaningful.

Discussion, You may want to include a more direct specific potential application or implication, such as quantify the level of fatigue by your device and algorithm. I also suggest highlighting your significance that there were few fatigue evaluation methods that can normalize to one’s own baseline or take personalized non-fatigue kinematic features into the algorithm account.

Reviewer 3 Report

The paper describes a methodology for estimating muscle fatigue in repeated spine movements using only data gathered from two inertial measurement units. A composite index is defined as a combination of ten measured variables, and its correlation to two different assessments of muscle fatigue is studied.

The authors conclude that the composite index can be a quite reliable indicator of muscle fatigue. Since the measurements required to compute the fatigue index don't require complex and expensive equipment, the method has a good applicability in real situations, which makes it very interesting.

The only minor concern, apart from the relatively small sample size, is the definition of the fatigue index itself. Although the authors mention in the Discussion that determining the importance of the individual variables would be interesting as a future work, it would be interesting to just compare the correlations obtained using each variable separately, thus giving the reader an idea of their relative importance.

Round 2

Reviewer 1 Report

The authors have clearly explained and revised my comments.